# Antifungal Activity of Earthworm Coelomic Fluid Obtained from *Eisenia andrei, Dendrobaena veneta* and *Allolobophora chlorotica* on Six Species of Phytopathogenic Fungi

**Sandra Ečimović** [1]**, Karolina Vrandečić** [2]**, Martina Kujavec** [1]**, Martina Žulj** [1]**, Jasenka Ćosić** [2]  **and Mirna Velki** [1,*]

[1] Department of Biology, Josip Juraj Strossmayer University of Osijek, Cara Hadrijana 8/A, 31000 Osijek, Croatia; sandra@biologija.unios.hr (S.E.); mkujavec@gmail.com (M.K.); martinna017@gmail.com (M.Ž.)

[2] Faculty of Agrobiotechnical Sciences Osijek, Josip Juraj Strossmayer University of Osijek, Vladimira Preloga 1, 31000 Osijek, Croatia; kvrandecic@pfos.hr (K.V.); jcosic@fazos.hr (J.Ć.)

[*] Correspondence: mirna.velki@gmail.com or mvelki@biologija.unios.hr

**Abstract:** The functioning of soil ecosystems greatly depends on the interactions occurring between soil biota communities. It is well known that earthworms are an important soil component that substantially affects its function, including their meaningful impact on the development of different phytopathogenic soil fungi. Phytopathogenic fungi are responsible for crop disease and cause great economic damage. It has previously been established that earthworms' coelomic fluid can suppress the growth of phytopathogenic fungi, but the exact molecular mechanism is unknown. The present study aimed at broadening the proof of this observed phenomenon by investigating the effects of the coelomic fluid extract of three different earthworm species (*Eisenia andrei, Dendrobaena veneta* and *Allolobophora chlorotica*) on the growth of six different phytopathogenic fungi species (*Berkeleyomyces basicola, Fusarium culmorum, Globisporangium irregulare, Rhizoctonia solani, Macrophomina phaseolina,* and *Sclerotinia sclerotiorum*). Coelomic fluid extract was obtained by electrostimulation or usage of extraction buffer (only in case of *A. chlorotica*) and prepared in three different concentrations by diluting the obtained coelomic fluid with physiological saline. The coelomic fluid extract of the three investigated earthworm species had an inhibitory effect on the growth of all six phytopathogenic fungi species. The greatest inhibitory effect was achieved with the *E. andrei* coelomic fluid extract reducing the growth of *R. solani* fungi. The findings of this research confirm the antifungal activity of coelomic fluid obtained from earthworm species belonging to different ecological categories and may be of potential use in crop protection against phytopathogenic fungi.

**Keywords:** crop protection; earthworm–fungi interaction; biocontrol; growth reduction



## 1. Introduction

There is a growing demand to increase the global agricultural production, which is challenged by many issues that result in losses of crop yield. For instance, phytopathogenic fungi cause significant losses in global crop production and can pose a serious threat to the world's food security [1]. *Berkeleyomyces basicola* (Berk & Broome) W.J. Nel, Z.W. de Beer, T.A. Duong & M.J. Wingf., *Fusarium culmorum* (Wm. G. Sm.) Sacc., *Globisporangium irregulare* (Buisman) Uzuhashi, Tojo & Kakish, *Rhizoctonia solani* J.G. Khün, *Macrophomina phaseolina* (Tassi) Goid and *Sclerotinia sclerotiorum* (Lib.) de Bary are soilborne plant pathogens with worldwide distribution and they cause a wide range of diseases with economically important yield losses. *S. sclerotiorum* and *M. phaseolina* infect over 500 cultivated and wild plant species [2,3], *R. solani* attacks more than 200 species [4]. *B. basicola* attacks over 120 plant species from at least 15 families [5], while *G. irregulare* attacks over 200 plant species [6]. *F. culmorum* is one of the most important *Fusarium* species and has a wide range of hosts, including corn, sorghum, small-grain cereals and many tame grass species

and weeds [7]. Additionally, it survives for a long period in plant debris and soil organic matter as microsclerotia, sclerotia, chlamydospores, oospores or mycelium [8]. Currently, conventional chemical fungicides play an important role in controlling soilborne diseases. On the other hand, it should be noted that excessive use of fungicides has significant negative consequences, including human health hazards, accumulation of residues in food, feed and soil, reduction of beneficial microorganisms, development of resistant fungal populations, and ecological disturbance [9,10]. To avoid the negative impact of fungicides on the environment, food and feed quality and quantity, and human health, there is a great need to develop new techniques to control plant pathogens.

Among terrestrial invertebrates, earthworms are directly and indirectly affected by the organisms present in the soil ecosystem, and they enter into symbiotic interactions with soil microorganisms, fungi and bacteria [11]. Because of their numerous positive activities in the soil, e.g., they modify chemical, physical and biochemical properties of the soil, earthworms are called "ecosystem engineers" [12]. By feeding on and ingesting a large amounts of fungal propagules earthworms can decrease the fungal biomass and consequently directly affect fungal populations [13]. By secreting the mucus and coelomic fluid, earthworms can indirectly negatively impact the fungal growth and contribute to a reduction of fungal biomass [14,15]. The coelomic fluid of earthworms comprises specific coelomic cells that play a significant role in defense reactions and immune responses [16]. Additionally, coelomic fluid itself possesses an abundance of bioactive substances that exhibit a variety of biological functions including bacteriostatic, proteolytic, cytolytic, antifungal and many other activities [17,18].

The results of earlier research have demonstrated the inhibitory effect of *E. andrei* and *D. venta* coelomic fluid extract on the growth *F. oxysporum* [19]. However, it is not known whether coelomocytes of other earthworm species have the same or similar effects on other fungal species. Therefore, the current goal was to assess the effect of coelomocyte extracts of three earthworm species on six species of phytopathogenic fungi (*B. basicola*, *F. culmorum*, *G. irregulare*, *M. phaseolina*, *R. solani* and *S. sclerotiorum*). In addition to two epigeic species, *E. andrei* (Bouché, 1972) [20] and *D. veneta* (Rosa, 1866) [21], the endogeic species *A. chlorotica* (Savigny, 1826) [22] was included in the study to determine whether earthworms of other ecological categories cause a growth reduction in agronomically important phytopathogenic fungi. The obtained results are of great importance in confirming the observed inhibitory potential of earthworm coelomic fluid on fungus growth and can provide a basis for the potential use of this phenomenon in disease management related to different phytopathogenic fungi.

## 2. Materials and Methods

### 2.1. Earthworms

Adult individuals with well-developed clitellae of the earthworm species *Eisenia andrei*, *Dendroabena veneta* and *Allobophora chlorotica* were purchased from local supplier. Before starting the experiment, earthworms were acclimatized for 2 weeks—they were placed in their breeding substrate and kept in the laboratory conditions. Prior to collection of coelomocytes, earthworms were rinsed with physiological saline solution and placed on moist filter paper for 24 h. This enabled voiding the gut therefore avoiding the contamination during the coelomocyte extraction.

### 2.2. Phytopathogenic Fungi

Phytopathogenic fungi of the species Berkeleyomyces basicola, Fusarium culmorum, Globisporangium irregulare, Rhizoctonia solani, Macrophomina phaseolina, and Sclerotinia sclerotiorum were provided from the culture collections Faculty of Agrobiotechnical Sciences Osijek, Croatia. All fungal cultures were maintained in Petri plates on potato dextrose agar (PDA, Difco, Detroit, Michigan) and kept in a growth chamber at $22 \pm 1\ °C$, with 12 h light/12 h dark regime.

### 2.3. Coelomocyte Isolation and Extract Preparation

Coelomocyte collection and extract preparation was conducted as described in [19] with some modifications. Since different number of coelomocytes (per earthworm) was extracted from different earthworm species, different total number of earthworms and/or different extraction method was applied in order to obtain enough coelomocytes to prepare all extracts for testing.

In short, ten earthworms of *E. andrei* species and five of *D. veneta* were washed, placed in physiological saline solution (4 mL) and stimulated by electricity current (5 V, 30 s). The whole procedure was performed on ice and the obtained extracts were kept on ice.

A sufficient amount of extract for all experimental treatments could not be obtained from *A. chlorotica* earthworm using electricity, so extrusion buffer was used to extract coelomocyte. Fifteen earthworms of *A. chlorotica* were washed and then placed into glass containing 8 mL of cold extrusion buffer. The extrusion buffer [23,24] used in the experiment was modified as described in [25]. Extruded coelomocytes were collected in Falcon tubes, centrifuged ($4000\times g$, 1 min, 4 °C), the pellets were resuspended in 4 mL physiological saline solution and the obtained extracts used in following procedures.

The total amount of coelomocytes per mL was determined using a cell counting chamber (Bürker-Türk). To determine the lowest effective concentration (i.e., causing the reduction in fungal growth), coelomic fluid extracts were diluted using physiological saline to desired concentrations. Preliminary tests have shown that concentrations of 2000, 3500 and 5000 coelomocytes/mL are required for *B. basicola F. culmorum*, *G. irregulare, M. phaseolina* and *R. solani* fungi, and 5000, 6000 and 7000 coelomocytes/mL for *S. sclerotiorum* fungus to reach significant effects. However, it was not possible to extract 6000 and 7000 cells/mL from the earthworm *A. chlorotica*, so concentrations of 2000, 3500 and 5000 coelomocytes/mL were used for treatment of *S. sclerotiorum* fungus with *A. chlorotica* coelomocyte extract. The control consisted only of physiological saline for *E. andrei* and *D. veneta*, while for the earthworm *A. chlorotica* control consisted of physiological saline and extrusion buffer.

### 2.4. Growth Inhibition Test

The growth inhibition test was conducted according to the methods previously described in [19]. Small Petri plates (ø5.8 cm) were filled with 5 mL of freshly prepared PDA and large Petri plates (ø9 cm) with 7.5 mL of PDA. After cooling, the culture medium was inoculated with fungus by placing a round (ø4 mm) sterile agar core of fungal culture in the middle of the Petri plate.

Immediately after fungal inoculation, each Petri dish was then treated with three different concentrations of coelomic fluid extract of three earthworm species separately, and controls were treated with physiological saline. For *B. basicola*, *F. culmorum*, *M. phaseolina* and *R. solani*, a volume of 1 mL was used for each concentration including control solution (small Petri plates ø5.8 cm), while for *S. sclerotiorum* and *G. irregulare* a volume of 2 mL of each concentration and control solution was applied due to the usage of a larger Petri plates (large Petri plates ø9 cm). The diameter of the aerial mycelium was measured every 24 h after application until the mycelium of the fungus outgrew the nutrient medium.

### 2.5. Data Analyses

Data analyses were performed using statistical software GraphPad Prism 5. Growth inhibition was estimated on the basis of the aerial mycelium diameter measured after the treatment and compared to control. The data were checked for normality by applying the Shapiro–Wilk test. The significance of the results was evaluated using one-way ANOVA followed by Dunnett's multiple comparison test to determine significant differences compared to control. The probability level for statistical significance was $p < 0.05$.

## 3. Results

The results from mycelium diameter measurements of *B. basicola* species treated with extracts of coelomic fluid of three earthworm species (*E. andrei*, *D. veneta* and *A. chlorotica*) are presented in Figure 1. *B. basicola*, unlike other fungi, had a slower growth, so mycelial growth was measured 24 h, 72 h, 6 days and 10 days after the treatment. The results showed significant growth inhibition 72 h, 6 and 10 days after exposure to coelomocyte extract of all earthworm species. The greatest reduction of mycelial growth (74.5%) was recorded 10 days after the treatment with extracts containing 3500 coelomocytes/mL of *D. veneta* earthworm species.

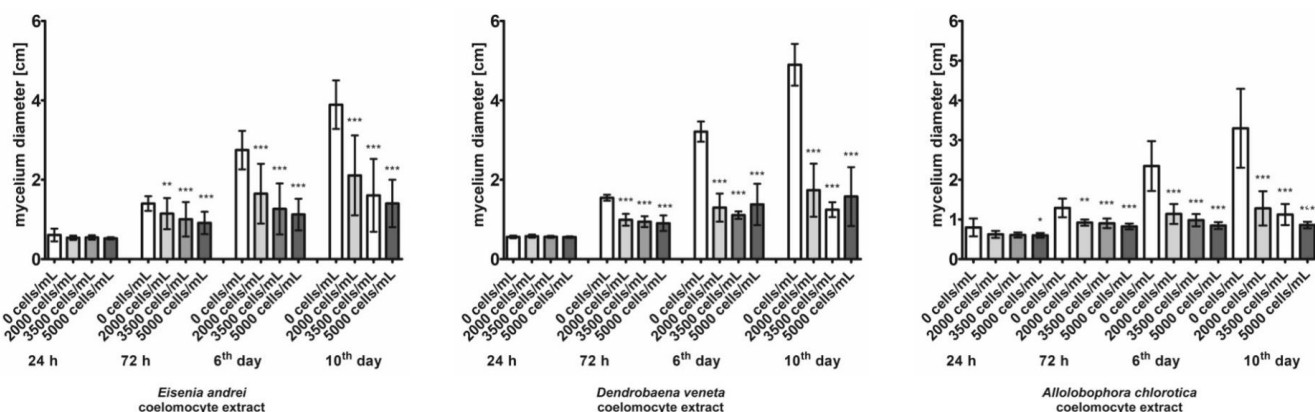

**Figure 1.** Growth (mean diameter ± SD) of *Berkeleyomyces basicola* mycelium 24 h, 48 h, 6 days and 10 days after treatment with *Eisenia andrei*, *Dendrobaena veneta* and *Allolobophora chlorotica* coelomic fluid extracts. Significant differences between control and groups treated with coelomocyte extracts are labeled with * ($p < 0.05$), ** ($p < 0.01$) and *** ($p < 0.001$).

According to the data presented in Figure 2, initial increase in mycelial growth *of F. culmorum* 24 h after the treatment is evident after exposure to all coelomocyte concentrations of *A. chlorotica* and two highest coelomocyte concentrations of *D. veneta*, whereas *E. andrei* coelomocyte extracts had no effect on fungi growth. Treatment with the highest concentration of *E. andrei* coelomocyte extract (5000 coelomocytes/mL) had the most pronounced effect on the mycelium of the fungus *F. culmorum* with 34.1% inhibition compared to the control 48 h after treatment. This inhibition of growth is also visible in Figure 3, where control fungi had a greater diameter of the aerial mycelium compared to fungi treated with *E. andrei* coelomocyte extract. Interestingly, 48 h after the treatment, a greater inhibitory effect was observed than after 72 h for this earthworm at all concentrations.

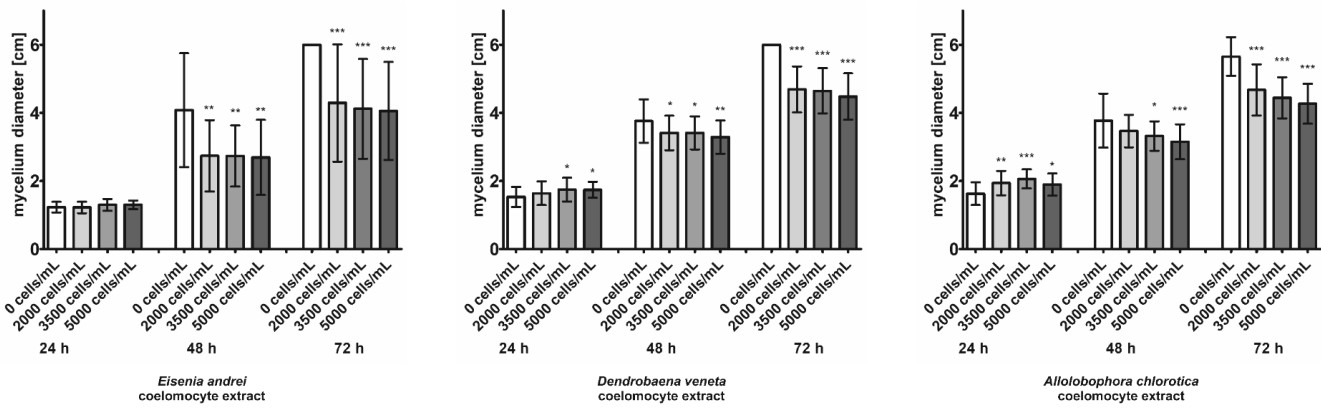

**Figure 2.** Growth (mean diameter ± SD) of *Fusarium culmorum* mycelium 24, 48 and 72 h after treatment with *Eisenia andrei*, *Dendrobaena veneta* and *Allolobophora chlorotica* coelomic fluid extracts. Significant differences between control and groups treated with coelomocyte extracts are labeled with * ($p < 0.05$), ** ($p < 0.01$) and *** ($p < 0.001$).

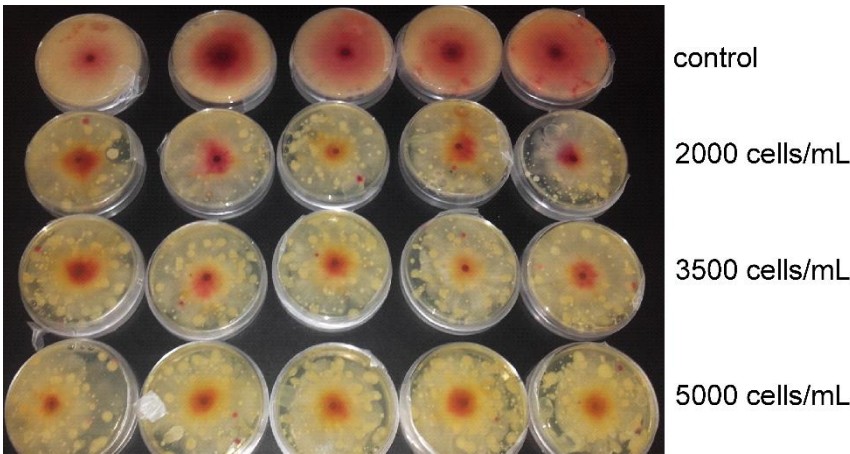

**Figure 3.** Petri dishes with *Fusarium culmorum* mycelium 48 h after treatment with *Eisenia andrei* coelomic fluid extract.

Coelomic extract of earthworm species *E. andrei*, *D. veneta* and *A. chlorotica* reduced the growth of fungus *M. phaseolina* (Figure 4). However, no significant differences were recorded between control and investigated coelomocyte concentrations 24 h after the treatment with *A. chlorotica* extracts. Treatment with extract containing 5000 coelomocytes/mL caused greatest growth reduction at all treatment periods (24, 48 and 72 h after treatment) in all earthworm species. The highest growth inhibition of *M. phaseolina* (44.7%) was recorded 72 h after treatment with the highest concentration of *A. chlorotica* extracts (5000 coelomocytes/mL). This inhibition of growth is presented in Figure 5, where greater diameter of the aerial mycelium in control fungi is clearly visible. Contrary to this, the lowest significant growth inhibition of *M. phaseolina* (13%) was recorded 24 h after treatment with the lowest concentration of *E. andrei* extracts (2000 coelomocytes/mL).

As in the case of *F. culmorum*, treatment with *D. veneta* coelomocyte extracts caused increase in mycelial growth of *G. irregulare* 24 h after the treatment, while 24 h after the treatment with *A. chlorotica* coelomic fluid extracts, significant growth inhibition was observed at the highest coelomocyte concentration applied (5000 coelomocytes/mL) (Figure 6). Growth reduction of *G. irregulare* was recorded 48 h after the treatment for all three earthworm species and all tested concentrations. The highest growth inhibition of *G. irregulare* (44.7%) was recorded 48 h after treatment with the highest concentration of *A. chlorotica* extracts (5000 coelomocytes/mL).

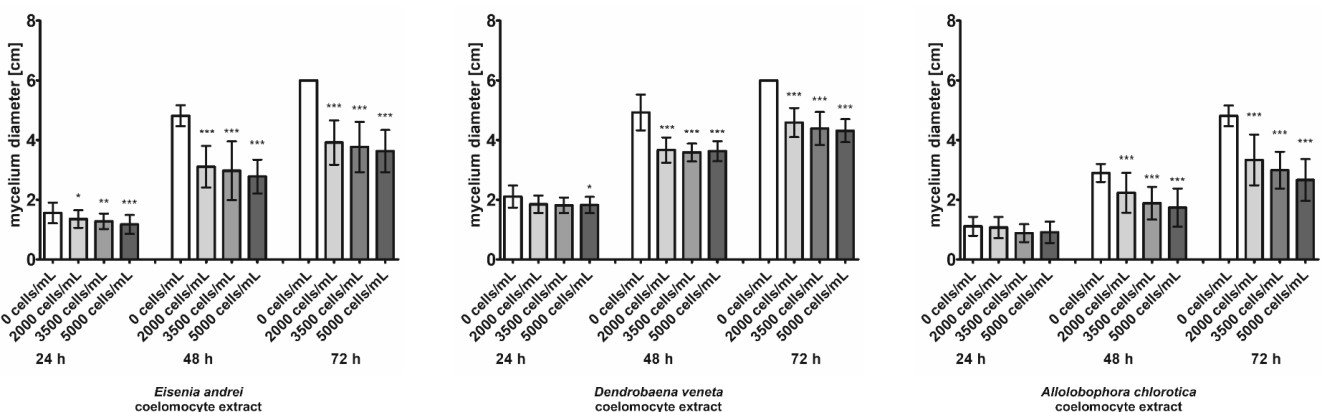

**Figure 4.** Growth (mean diameter ± SD) of *Macrophomina phaseolina* mycelium 24, 48 and 72 h after treatment with *Eisenia andrei*, *Dendrobaena veneta* and *Allolobophora chlorotica* coelomic fluid extracts. Significant differences between control and groups treated with coelomocyte extracts are labeled with * ($p < 0.05$), ** ($p < 0.01$) and *** ($p < 0.001$).

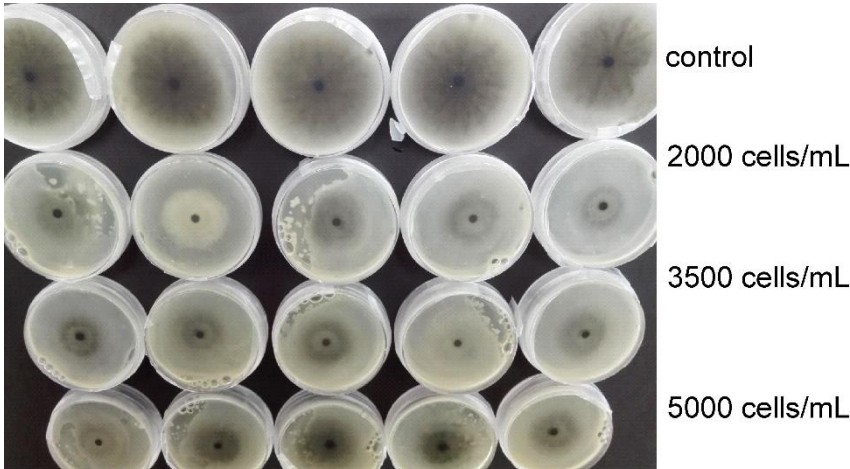

**Figure 5.** Petri dishes with *Macrophomina phaseolina* mycelium 72 h after treatment with *Allolobophora chlorotica* coelomic fluid extract.

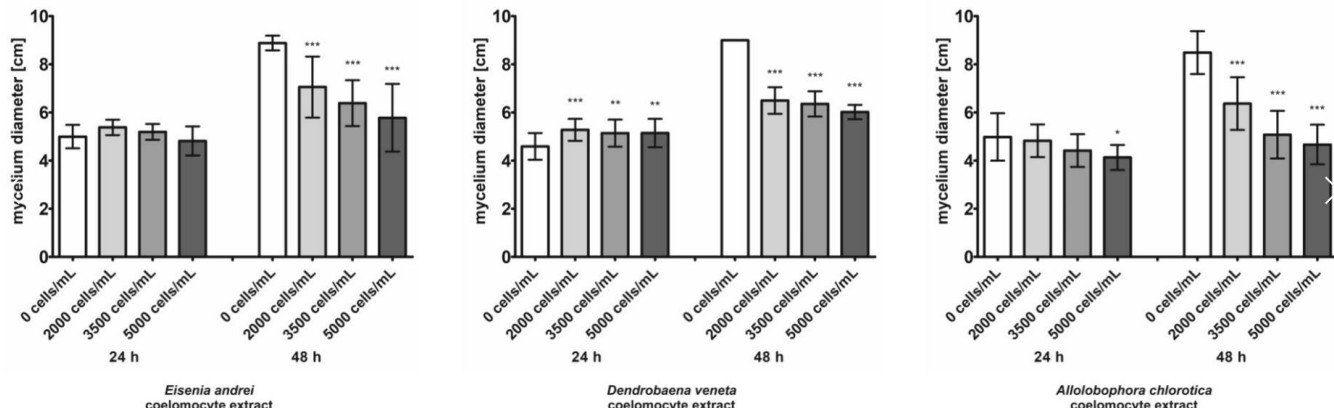

**Figure 6.** Growth (mean diameter ± SD) of *Globisporangium irregulare* mycelium 24 h and 48 h after treatment with *Eisenia andrei*, *Dendrobaena veneta* and *Allolobophora chlorotica* coelomic fluid extracts. Significant differences between control and groups treated with coelomocyte extracts are labeled with * ($p < 0.05$), ** ($p < 0.01$) and *** ($p < 0.001$).

Treatment of *R. solani* with *E. andrei* and *D. veneta* coelomocyte extract caused significant differences between all extract concentrations compared to control at all time periods of mycelial diameter measurement (Figure 7). However, 24 h after the treatment of *R. solani* with different concentrations of *A. chlorotica* coelomic fluid extracts significant differences in growth were not observed. The highest growth inhibition of *R. solani* (83.7%) was recorded 96 h after treatment with the highest concentration of *E. andrei* extracts (5000 coelomocytes/mL), which is also the highest percentage of inhibition recorded in this investigation.

Results of *S. sclerotiorum* fungal mycelium growth treated with *E. andrei* extract showed significant differences between controls and concentrations of 5000, 6000 and 7000 coelomocytes/mL 48 and 72 h after treatment (Figure 8). The results of measuring the diameter of *S. sclerotiorum* mycelium after treatment with *D. veneta* coelomocyte extract showed significant growth reduction 24 h after treatment with 7000 coelomocyte/mL extract and 48 and 72 h after treatment at all concentrations (5000, 6000 and 7000 coelomocytes/mL). Significant differences were noticed between control and all tested concentrations (5000, 6000 and 7000 coelomocyte/mL) of *A. chlorotica* extracts at all times (24, 48 and 72 h after treatment) of measuring mycelium diameters of *S. sclerotiorum*. The highest growth inhibition (74%) of *S. slerotiorum* caused by *D. veneta* extracts was recorded 72 h after the treatment with extract concentration of 7000 coelomocytes/mL.

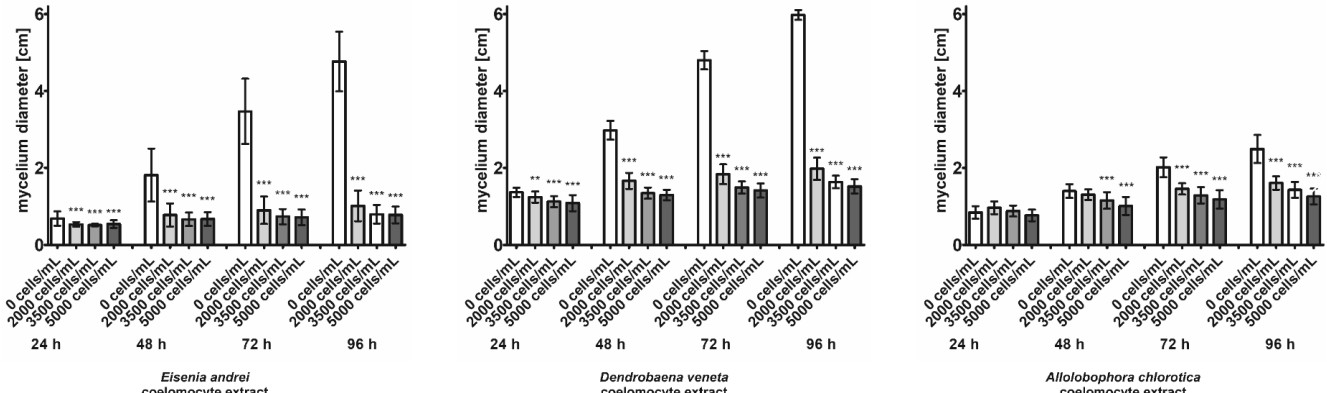

**Figure 7.** Growth (mean diameter ± SD) of *Rhizoctonia solani* mycelium 24, 48, 72 and 96 h after treatment with *Eisenia andrei*, *Dendrobaena veneta* and *Allolobophora chlorotica* coelomic fluid extracts. Significant differences between control and groups treated with coelomocyte extracts are labeled with ** ($p < 0.01$) and *** ($p < 0.001$).

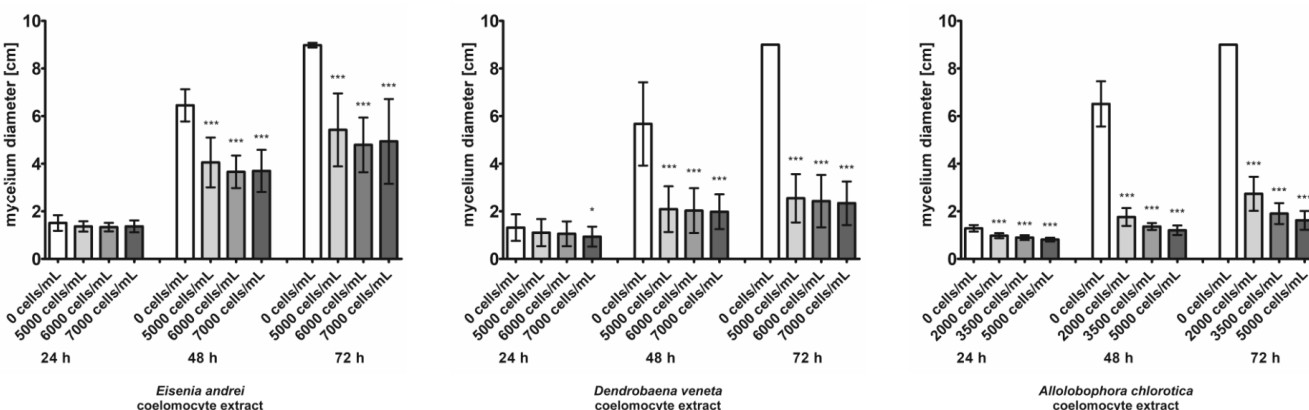

**Figure 8.** Growth (mean diameter ± SD) of *Sclerotinia sclerotiorum* mycelium 24, 48 and 72 h after treatment with *Eisenia andrei*, *Dendrobaena veneta* and *Allolobophora chlorotica* coelomic fluid extracts. Significant differences between control and groups treated with coelomocyte extracts are labeled with * ($p < 0.05$), ** ($p < 0.01$) and *** ($p < 0.001$).

## 4. Discussion

Previous research [19] has demonstrated the inhibitory effect of coelomic fluid extract of two earthworm species on the phytopathogenic fungi and considering the importance of such finding, as well as its potential practical application, the present study expanded this research and assessed effects of coelomic fluid extracts of three earthworm species (*E. andrei, D. veneta* and *A. chlorotica*) on growth of six phytopathogenic fungi (*B. basicola, F. culmorum, G. irregulare R. solani, M. phaseolina*, and *S. sclerotiorum*).

It is known that earthworms, in addition to their direct influence through the digestion of mycelium and fungal spores, can also indirectly reduce the survival of phytopathogenic fungi [13,26]. For example, anecic earthworm species such as *Lumbricus terrestris*, which take fresh leaf cover from the surface and carry it deep into the soil, can reduce populations of phytopathogenic fungi that feed on saprophytic fresh leaf cover such as *Fusarium spp.*, taking food to places inaccessible to them [27]. Moreover, it has been shown that bacterium *Serratia marcescens* isolated from the cuticle of earthworms strongly inhibited the germination of *F. proliferatum* conidia and mycelial growth [28]. Although previous research has shown that earthworms can affect soil fungi both directly and indirectly [13,26,29,30], the investigations of the effect of coelomic fluid on fungi are rare. Coelomic fluid of earthworms contains coelomocytes, various bioactive molecules such as lysenins, fetidins, lumbricin, cytolytic protein eiseniapore, eritrocytolytic proteins and coelomic cytolytic factor (CCF-1), which plays an important role in the immune system [31–34]. In addition, it

contains many enzymes (proteases, lysozymes, metalloenzymes, fibrinolytic enzymes and polysaccharides, antimicrobial proteins, nutrients, etc.) [35]. Coelomocytes have numerous functions in the body, for example phagocytosis, they are involved in wound healing, cytotoxicity, inflammation, encapsulation of foreign substances, detoxification, coelomic fluid coagulation and other [36–40]. Considering the composition of the coelomic fluid and the presence of diverse bioactive molecules, the coelomic fluid has potential to act on microorganisms in the surroundings of earthworms.

As mentioned previously, [19] demonstrated that coelomocyte extracts of earthworms *D. veneta* and *E. andrei* inhibit the growth of phytopathogenic fungus *Fusarium oxysporum*. The question arose as to whether this observed inhibition is specific to the tested earthworm and fungus species, or whether it can be observed in the case of other earthworm species and phytopathogenic fungi. The results obtained in this study showed that coelomocyte extract of all three investigated species of earthworm, belonging to the epigeic and endogeic ecological categories, had an inhibitory effect on the growth of six tested phytopathogenic fungi. The greatest growth inhibition (83.7%) in relation to all extracts, but also to all investigated fungi, was achieved by application of the coelomocyte extract of the species *E. andrei* in treatment of fungus *R. solani*, whereas the weakest inhibitory effect was observed in treatment of *F. culmorum* with extracts of all earthworm species. A high percentage of inhibition was also obtained with coelomocyte extracts of *D. veneta* in the treatment of *B. elegans* (74.6%), as well as in the treatment of *B. elegans* with *A. chlorotica* earthworm extract (73.9%). In almost all cases, the highest concentration of applied coelomocyte extract caused the greatest growth inhibition. The most pronounced differences between the control and treated groups were obtained 72 h after the treatment, except for the fungus *B. elegans*, where the strongest effect was observed after 10 days, whereas in the first measurement (24 h after treatment), these differences were less pronounced for all treatments. These results are in good agreement with the previously published work where coelomocyte extracts of earthworms *D. veneta* and *E. fetida* significantly reduced the growth of phytopathogenic fungus *F. oxysporum* [19]. The obtained results obviously confirmed the previously observed inhibitory effect of coelomic fluid extract on the growth of phytopathogenic fungi, with emphasis on the broadened proof of this effect in the context of usage of three different earthworm species and six phytopathogenic fungi.

Several studies also demonstrated antifungal properties of coelomic fluid and mucus of earthworms [41–48]; however, the exact mechanism by which coelomic fluid causes inhibition of fungal growth is not yet known. It is likely to be associated with hemolytic, antibacterial, cytotoxic, and proteolytic functions of coelomic fluid and coelomocytes [31]. The bioactive molecules in the coelomic fluid, lysenin and lumbricin I, could be responsible for fungal growth inhibition. Studies showed that lumbricin I exhibits antimicrobial activity against broad spectrum of microorganism and fungi [36]. Lysenin could also be involved in fungal cell lysis; by binding to sphingomyelin, it forms pores, and this interaction leads to cytotoxicity [31]. Fungi are eukaryotes and, as with all eukaryotes, their membranes contain sphingolipids [31,49]. If lysenin can bind to some of the sphingolipids, it would be possible to lysenin causes lysis of fungal membranes and cells. A protein-carbohydrate fraction was isolated from the coelomic fluid of earthworm *D. veneta*, which showed antifungal activity against *Candida albicans* and effectively destroyed their cells [35]. It is possible that multiple factors and active molecules are involved in this mechanism of growth inhibition of phytopathogenic fungi. In addition, there is a possibility that antibacterial peptide such as ECPF [50] isolated from coelomic fluid of earthworm *E. fetida*, which exerts hemolytic, antibacterial and antitumor activities, is also involved in this mechanism. There are also several antimicrobial peptides, such as F-1 and F-2, isolated and purified from *E. andrei*, which play an important role when the immune system is invaded by external pathogens [51].

## 5. Conclusions

The large-scale use of synthetic fungicides for control of plant diseases causes soil, air, food and water contamination. In addition, due to their widespread application, there is a tendency of resistance development in fungal populations. Therefore, other options for fungus control have to be considered. Due to the harmful effects of fungicides, it is necessary to investigate possibilities of controlling diseases caused by phytopathogenic fungi that will not adversely affect beneficial organisms in the environment, and the results obtained in the present study show the potential use of earthworm coelomic fluid extract for this purpose. Namely, it is evident that coelomic fluid of different species of earthworms causes an inhibitory effect on the growth of different phytopathogenic fungus species, and consequently has potential practical applications in crop protection against phytopathogenic fungi.

The observed inhibitory effect of coelomic fluid extract of several earthworm species on growth of different phytopathogenic fungi species is an important finding. The inhibition strength was unequal in different fungi species so the following research should focus on elucidation of the mechanisms by which coelomic fluid and coelomocytes affect growth inhibition. Additionally, it would be desirable to carry out complete biochemical characterization of the coelomic fluid extracts of different earthworm species in order to determine exact composition of bioactive molecules. Knowledge of the biochemical composition of coelomic fluid can contribute to better and targeted selection of a particular compound of coelomic fluid for the purpose of biocontrol of phytopathogenic fungi.

**Author Contributions:** Conceptualization, S.E. and M.V., conduction of the experiments, M.K. and M.Ž., supervision, S.E., K.V., J.Ć., M.V., data analysis, M.V., draft preparation, S.E., manuscript finalization, S.E., K.V., M.K., M.Ž., J.Ć., M.V. All authors have read and agreed to the published version of the manuscript.

**Funding:** This research received no external funding.

**Institutional Review Board Statement:** Not applicable.

**Informed Consent Statement:** Not applicable.

**Data Availability Statement:** Data available upon request.

**Conflicts of Interest:** The authors declare no conflict of interest.

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
