# Peer review of "Antifungal Activity of Earthworm Coelomic Fluid Obtained from Eisenia andrei, Dendrobaena veneta and Allolobophora chlorotica on Six Species of Phytopathogenic Fungi"

_environments, doi:10.3390/environments8100102_

Round 1
Reviewer 1 Report
The manuscript reports the results of a laboratory experiment to assess the effect, if any, of coelomic fluid from three different earthworm species on six phytopathogenic fungal species.
The subject of the manuscript is of current scientific interest and aligns well with current societal push for sustainable methods to underpin food production. Overall, the experimental design appears appropriate though some methodological information is missing (see detailed comments). Unfortunately there is a fundamental problem with the Results section in that potentially the authors are unfamiliar with the concept of statistical difference and when values can be considered as different. Thus, much of what is stated in the Results needs to be rewritten for accuracy to reflect the outcome of the statistical analysis.
In addition to the above major point, I would ask the authors to reflect upon the following:
Line 10: "proper" what do the authors mean by this?
Keywords: "earthworms", "coelomic fluid extract" and "fungi" are in various forms part of the title and thus are redundant as keywords, please replace
Line 43: what do the authors mean by "tame"?
Lines 71-77: requires a reference to earthworm functional groups e.g. Bouche, Botinelli
Lines 95 & 99: why do the earthworm numbers (10, 5 and 15) change?
Line 98: please be accurate, what is "A sufficient amount"?
Lines 142-145: rewrite for simplicity stating the key result rather than taking several lines
Lines 148-152: please be accurate - at 24h all concentrations for A. chlorotica had an increase; 3500 and 5000 for D. veneta had an increase; there was no change for E. andrei
Lines 163-165: this statement is not supported statistically, i.e. there are no differences
Line 166: again there is no statistical support for E. andrei
Line 169: incorrect, only the 5000 concentration is statistically supported for A. chlorotica
Lines 174-176: there is no trend as with time the reduction is similar across all concentrations
Line 174: Fig 6 should be Fig 5
Lines 186-188: but 6000 was not statistically different to either 5000 or 7000
Lines 191-194: this sentence is incorrect based on the output of the statistical analysis
Lines 285-304: speculative text
Figure 1: if the outcome of the statistical analysis is reported as p<0.05 (and see lines 135-137), how can the authors provide statistical differences of three different statistical levels as noted by three different asterisks?
Figure 3 is not cited in the text
References: species names are not italicised though this may be a journal specific format, the occasional incorrect (non standard ISI) journal abbreviation, the occasional page numbering issue (for example, lines 368-369)
Author Response
General comments: The manuscript reports the results of a laboratory experiment to assess the effect, if any, of coelomic fluid from three different earthworm species on six phytopathogenic fungal species. The subject of the manuscript is of current scientific interest and aligns well with current societal push for sustainable methods to underpin food production. Overall, the experimental design appears appropriate though some methodological information is missing (see detailed comments). Unfortunately there is a fundamental problem with the Results section in that potentially the authors are unfamiliar with the concept of statistical difference and when values can be considered as different. Thus, much of what is stated in the Results needs to be rewritten for accuracy to reflect the outcome of the statistical analysis.
We thank this reviewer for comments which helped to us improve our manuscript. We have accepted all suggestions and made according changes. Our replies are in italic.
*Line 10: "proper" what do the authors mean by this?
We meant that everything is functioning as it should. We now see that this word is redundant and have therefore removed it.
*Keywords: "earthworms", "coelomic fluid extract" and "fungi" are in various forms part of the title and thus are redundant as keywords, please replace.
Replaced accordingly.
*Line 43: what do the authors mean by "tame"?
“Tame” grass species are introduced (non-native) grass species. (“Tame pastures are cultivated fields planted with introduced (non-native) grass cultivars with the multiple purposes of providing livestock grazing forage to improve animal nutrition and health, balance forage (Jacobs and Siddoway, 2009)”).
*Lines 71-77: requires a reference to earthworm functional groups e.g. Bouche, Botinelli.
References were added.
*Lines 95 & 99: why do the earthworm numbers (10, 5 and 15) change?
Due to the difference in the amount of coelomocytes that could be obtained from each species. Namely, in order to be able to compare the results between species we had to have same concentration of coelomocytes in prepared extracts. And for some species we had to increase the number of earthworms (or even change method as described for A. chlorotica) to obtain enough coelomocytes. We have added short explanation in order to make that clear.
*Line 98: please be accurate, what is "A sufficient amount"?
This relates to previous question and explanation. Namely, a different number of coelomocytes (per earthworm) was extracted from different earthworm species, so we had to change number of earthworms and/or method in order to obtain enough coelomocytes to prepare all extracts for testing (in different concentration and sufficient volume for all tests). We have specified “A sufficient amount” to “A sufficient amount of extracts for all experimental treatments”.
*Lines 142-145: rewrite for simplicity stating the key result rather than taking several lines
Rewritten as requested.
*Lines 148-152: please be accurate - at 24h all concentrations for A. chlorotica had an increase; 3500 and 5000 for D. veneta had an increase; there was no change for E. andrei
We apologize for inaccurate description, we have corrected it as suggested.
*Lines 163-165: this statement is not supported statistically, i.e. there are no differences.
We agree. Here we only looked at % of growth inhibition and overlooked the fact that this decrease was not significant. We apologize for this mistake and we have corrected it.
*Line 166: again there is no statistical support for E. andrei.
We apologize for the mistake. It has been corrected.
*Line 169: incorrect, only the 5000 concentration is statistically supported for A. chlorotica.
We apologize for the mistake. It has been corrected.
*Lines 174-176: there is no trend as with time the reduction is similar across all concentrations.
We agree that the reduction is similar, but still in all cases was at least a bit smaller at increasing concentrations. Since the difference is small, to avoid misunderstanding, we have removed this sentence.
*Line 174: Fig 6 should be Fig 5.
Corrected.
*Lines 186-188: but 6000 was not statistically different to either 5000 or 7000.
We agree. We have removed this sentence.
*Lines 191-194: this sentence is incorrect based on the output of the statistical analysis.
We agree that the part of the trend is not correct. We have removed that sentence. The other sentence, regarding the specific significances, is correct.
*Lines 285-304: speculative text.
All statements given here are supported by references. And since the mechanism of the observed phenomenon is not known, we actually had to speculate about possible mechanisms. For sure, in the future studies, the actual mechanisms of the observed growth inhibition should be assessed.
*Figure 1: if the outcome of the statistical analysis is reported as p<0.05 (and see lines 135-137), how can the authors provide statistical differences of three different statistical levels as noted by three different asterisks?
As stated in M&M section, the probability level for statistical significance was p < 0.05. That means that p value had to be lower than 0.05 in order to consider the difference significant. However, we also made labels in cases of even “stronger” significance, i.e. when p values were lower than 0.01 (two asterisks) and 0.001 (three asterisks). We accidentally omitted that labels in the figure captions. We apologize for the mistake, we have now added them.
*Figure 3 is not cited in the text.
It is cited in line 157, following sentence: Coelomic extract of earthworm species E. andrei, D. veneta and A. chlorotica reduced the growth of fungus M. phaseolina (Fig. 3).
*References: species names are not italicised though this may be a journal specific format, the occasional incorrect (non standard ISI) journal abbreviation, the occasional page numbering issue (for example, lines 368-369)
We apologize, we have made appropriate corrections.
Reviewer 2 Report
Journal: Environments
Manuscript ID: environments-1363333
Type of manuscript: Article
Title: Antifungal activity of earthworm coelomic fluid obtained from Eisenia andrei, Dendrobaena veneta and Allolobophora chlorotica on six species of phytopathogenic fungi
Authors: Sandra Ečimović, Karolina Vrandečić, Martina Kujavec, Martina Žulj, Jasenka Ćosić, Mirna Velki * Feature Papers in Environments in 2021
General comments:
The submitted manuscript describes the antifungal properties of coelomic fluid from 3 earthworm species. The study is nicely done, the whole manuscript is clearly and concisely written. The English are without big issues and the structure of the article is correct. A coelomic fluid of earthworms, especially of E. andrei species, is a very active liquid and has many bioactive properties like antimicrobial, hemolytic and cytolytic properties. The anifungal activity of coelomic fluie has not been a much-researched topic so far and it is very useful to continue this topic. Overall, after some necessary corrections, I consider this manuscript to be suitable for publication in Environments.
Major comments:
1) How long did authors acclimatized the earthworms in the laboratory and in what substrate? It should be added to the manuscript.
2) Authors used extracted coelomocytes in coelomic fluid for the fungi growth inhibition. Why did they used the coelomocytes and not coelomic fluid only or cellular lysate? It is very likely that active substances are mainly in coelomic fluid or inside the cells.
3) Which types of cells did they calculate? Earthworms have 3 different cell subpopulations – chloragocytes, hyaline and granular amoebocytes. They are clearly distinguishable, but chloragocytes are very fragile and do not have an immune function. On the other hand, these cells are the main producers of lysenin…
4) It would be very suitable to add some pictures of fungal mycelium on dishes in time.
5) Results: Fig. 2 Authors described that the highest inhibition was in the case of E. andrei coelomocytes (5000 coel/ml) 48h after the treatment. But from the figure, it doesn't seem so. There is greater inhibition 72 h after the treatment as is further indicated by 3 asterisks.
6) The reference to the results from Figure 5 should be in the text before Figure 6.
7) Fig. 4 . Why there is not 72 h interval in the graph when the results are mentioned in the test?
Minor comments:
-l.49 – the parallel structure is not maintained here., beter to use „development“.
-l. 53-56 – the parallel structure is not maintained here. The sentence should be rewritten.
Author Response
General comment: The submitted manuscript describes the antifungal properties of coelomic fluid from 3 earthworm species. The study is nicely done, the whole manuscript is clearly and concisely written. The English are without big issues and the structure of the article is correct. A coelomic fluid of earthworms, especially of E. andrei species, is a very active liquid and has many bioactive properties like antimicrobial, hemolytic and cytolytic properties. The anifungal activity of coelomic fluie has not been a much-researched topic so far and it is very useful to continue this topic. Overall, after some necessary corrections, I consider this manuscript to be suitable for publication in Environments.
We thank very much to this reviewer for the comments that have helped us to improve the manuscript. We have accepted or explained all recommendations and improved the version of manuscript accordingly. Our replies are in italics.
Major comments:
*1) How long did authors acclimatized the earthworms in the laboratory and in what substrate? It should be added to the manuscript.
Earthworms were acclimatized for 2 weeks in breeding substrate obtained from the supplier. We have added this information in the manuscript.
*2) Authors used extracted coelomocytes in coelomic fluid for the fungi growth inhibition. Why did they used the coelomocytes and not coelomic fluid only or cellular lysate? It is very likely that active substances are mainly in coelomic fluid or inside the cells.
Since we had three different earthworm species and we wanted to compare the effects of their coelomocyte extracts, we had to somehow “standardize” extracts. Since the extraction is not always equally effective, we could not just extract the coelomic fluid and compare between species. So in order to have some reference point for comparison we chose to count the number of coelomocytes and prepare extracts of different species with the same cell concentration. We agree that it would be interesting to have coelomic fluid or lysate to determine the mechanism of the observed phenomenon due to active substances that may be present. But this was not the aim of the present study. However, due to the observed results, it is something that should be definitively considered for future studies.
*3) Which types of cells did they calculate? Earthworms have 3 different cell subpopulations – chloragocytes, hyaline and granular amoebocytes. They are clearly distinguishable, but chloragocytes are very fragile and do not have an immune function. On the other hand, these cells are the main producers of lysenin…
We included all cells. We counted all coelomocytes and then expressed the concentration of the extract in number of cells per mL. Since we had different earthworm species, which may have different number of particular coelomocytes, for this study the best option was to include them all. Since we observed different strength of growth inhibition for different earthworm species, it would be interesting to investigate if this is related to different number of particular type of coelomocytes present in different earthworm species. However, this was beyond this study and should be a topic for future investigations.
*4) It would be very suitable to add some pictures of fungal mycelium on dishes in time.
We agree. We have added the pictures.
*5) Results: Fig. 2 Authors described that the highest inhibition was in the case of E. andrei coelomocytes (5000 coel/ml) 48h after the treatment. But from the figure, it doesn't seem so. There is greater inhibition 72 h after the treatment as is further indicated by 3 asterisks.
Here by the “highest inhibition” is meant the highest percentage of the inhibition and not the significance. Namely, after 72 h all control fungi overgrew the Petri dish and this is why there is no variation in the mycelium diameter. Because of that the significance is “greater” (i.e. three asterisks) compared to 48 h where due to present variability in the control the significance is “lower” (i.e. two asterisks). However, the % of the growth inhibition is higher after 48 h compared to 72 h.
*6) The reference to the results from Figure 5 should be in the text before Figure 6.
Corrected.
*7) Fig. 4. Why there is not 72 h interval in the graph when the results are mentioned in the test?
We apologize for the mistake, it should be 48 h (not 72 h) in the text. We have corrected that.
Minor comments:
*-l.49 – the parallel structure is not maintained here., beter to use „development“.
Corrected.
*-l. 53-56 – the parallel structure is not maintained here. The sentence should be rewritten.
Corrected.
Round 2
Reviewer 1 Report
I thank the authors for addressing the points that I outlined in my original review. Remaining are the following minor points which I would ask the authors to consider:
Line 43: Fusarium should be italicised
Line 67 and elsewhere: the use of "proved" should be avoided unless the authors are 100% sure that the results they refer to specifically are related to a postulated hypothesis. If there was not hypothesis being addressed such as in the current manuscript then nothing can be "proved" as the word has a very specific meaning. I would sugeest replacing "proved" with perhaps "demonstrated"
Line 153 and elsewhere: there is no need to have data expressed to the resolution of two decimal places, one is sufficient
Figure 6: the panel for D. veneta has the y-axis in bold, the others in the Figure are not
Figure 7: please revise the legend as there is no * displayed
References: still several issues remaining including inconsistent formatting of article titles (e.g. reference #8); incorrect journal abbreviations used (e.g. reference #17)
Author Response
Line 43: Fusarium should be italicised
Corrected.
Line 67 and elsewhere: the use of "proved" should be avoided unless the authors are 100% sure that the results they refer to specifically are related to a postulated hypothesis. If there was not hypothesis being addressed such as in the current manuscript then nothing can be "proved" as the word has a very specific meaning. I would sugeest replacing "proved" with perhaps "demonstrated"
Changed as suggested.
Line 153 and elsewhere: there is no need to have data expressed to the resolution of two decimal places, one is sufficient
Changed as suggested.
Figure 6: the panel for D. veneta has the y-axis in bold, the others in the Figure are not
We could not find this. All figures are identically formatted.
Figure 7: please revise the legend as there is no * displayed
Revised.
References: still several issues remaining including inconsistent formatting of article titles (e.g. reference #8); incorrect journal abbreviations used (e.g. reference #17)
Corrected.
